CALT-TH-2020-035

# On Exotic Consistent Anomalies in (1+1)$d$:
# A Ghost Story

Chi-Ming Chang[a], Ying-Hsuan Lin[b,c]

[a]*Yau Mathematical Sciences Center, Tsinghua University, Beijing, 10084, China*

[b]*Walter Burke Institute for Theoretical Physics,*
*California Institute of Technology, Pasadena, CA 91125, USA*

[c]*Jefferson Physical Laboratory, Harvard University, Cambridge, MA 02138, USA*

`cmchang@tsinghua.edu.cn, yinhslin@gmail.com`

## Abstract

We revisit 't Hooft anomalies in $(1+1)d$ non-spin quantum field theory, starting from the consistency and locality conditions, and find that consistent U(1) and gravitational anomalies cannot always be canceled by properly quantized $(2+1)d$ classical Chern-Simons actions. On the one hand, we prove that certain exotic anomalies can only be realized by non-reflection-positive or non-compact theories; on the other hand, without insisting on reflection-positivity, the exotic anomalies present a caveat to the inflow paradigm. For the mixed U(1) gravitational anomaly, we propose an inflow mechanism involving a mixed U(1)×SO(2) classical Chern-Simons action with a boundary condition that matches the SO(2) gauge field with the $(1+1)d$ spin connection. Furthermore, we show that this mixed anomaly gives rise to an isotopy anomaly of U(1) topological defect lines. The isotopy anomaly can be canceled by an extrinsic curvature improvement term, but at the cost of creating a periodicity anomaly. We comment on a subtlety regarding the anomaly of finite subgroups of U(1), and end with a survey of the holomorphic $bc$ ghost system which realizes all the exotic consistent anomalies.

# 1   Introduction

't Hooft anomaly is a controlled breaking of symmetries in quantum field theory (QFT). Let $\Phi$ collectively denote the background gauge fields and metric, and $\Lambda$ collectively denote diffeomorphisms and background gauge transformations. Under $\Lambda$, the partition function on $\Phi$ transforms as

$$Z[\Phi^\Lambda] = Z[\Phi]\, e^{i\alpha[\Phi,\Lambda]}, \tag{1.1}$$

The anomalous phase $\alpha[\Phi, \Lambda]$ is a functional that satisfies the consistency and locality conditions. Consistency — or finite Wess-Zumino consistency [1] — of an 't Hooft anomaly requires the background gauge transformation (1.1) to respect the group multiplication law, which amounts to the commutativity of the diagram

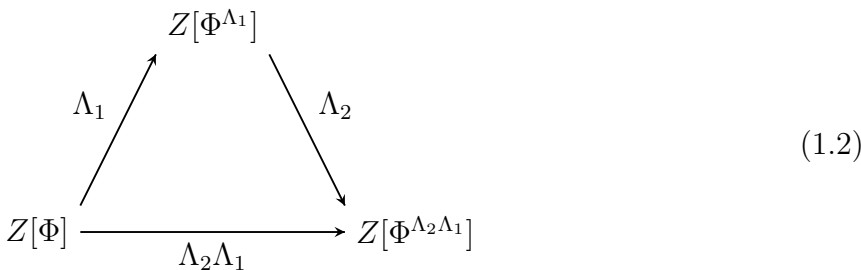

$$(1.2)$$

The anomalous phases generated by the two routes differ by $2\pi\mathbb{Z}$. Locality of an 't Hooft anomaly is expected because anomaly is a short distance effect, *i.e.* it originates in the ultraviolet. The consistency and locality conditions led to the old cohomological classification of perturbative anomalies – the 't Hooft anomaly of a semi-simple Lie algebra $\mathcal{G}$ in $D$ spacetime dimensions is classified by the Lie algebra cohomology $H^{D+1}(\mathcal{G}, \mathbb{R})$ through the descent equations [2–13].

A more modern perspective on 't Hooft anomalies is the *inflow paradigm*: a $D$-dimensional anomalous QFT should be viewed as the boundary theory of a $(D + 1)$-dimensional bulk classical action, also called a symmetry protected topological phase or an invertible field theory, such that the coupled system exhibits no anomaly [14–28]. From this perspective, the classification of boundary 't Hooft anomalies amounts to the classification of bulk classical actions. One recent triumph has been the classification of *reflection-positive* invertible topological field theories in $D + 1$ spacetime dimensions by cobordism groups [20, 25, 28].[1]

For a discrete internal symmetry group $G$ in a $(1+1)d$ non-spin QFT, the inflow paradigm suggests that the 't Hooft anomalies have the same $H^3(G, \mathrm{U}(1))$ classification as the $(2+1)d$ Dijkgraaf-Witten theories [29]. The same classification can also be deduced from a purely $(1+1)d$ perspective [30, 31].[2] According to the inflow paradigm, the chiral central charge $c_- \equiv c - \bar{c}$ of a $(1+1)d$ non-spin CFT must be a multiple of eight, because only then can the gravitational anomaly be canceled by a properly quantized $(2+1)d$ gravitational Chern-Simons action. Similarly, the level $k_- \equiv k - \bar{k}$ of a $\mathrm{U}(1)$ internal symmetry in a non-spin CFT must be an even integer for the $\mathrm{U}(1)$ anomaly to be canceled by a $(2+1)d$ $\mathrm{U}(1)$ Chern-Simons.

---

[1] Reflection positivity of a QFT in Euclidean spacetime is equivalent to the unitarity of time evolutions in Lorentzian spacetime. However, in this paper we always call this property reflection-positivity, to avoid confusion of QFT unitarity with the unitarity of symmetry representations.

[2] The $(1+1)d$ classification is achieved by the pentagon identity, which arises as the consistency condition for the fusion category of symmetry defect lines, or equivalently from the finite Wess-Zumino consistency condition applied to patch-wise background gauge transformations.

The above quantization conditions are violated by the holomorphic $bc$ ghost system. Recall that $b$ and $c$ are left-moving anti-commuting free fields with weights $\lambda$ and $1 - \lambda$. For integer $\lambda$, the holomorphic $bc$ ghost system is a non-spin CFT, but has $c_- = 1 - 3(2\lambda - 1)^2 \in 2\mathbb{Z}$ and $k_- = 1$, suggesting that the gravitational and U(1) anomalies cannot be canceled by inflow of familiar Chern-Simons actions. On the other hand, the consistency and locality conditions lead to weaker quantization conditions $c_- \in 2\mathbb{Z}$ and $k_- \in \mathbb{Z}$ that are precisely satisfied by the holomorphic $bc$ ghost system.

The $bc$ ghost system has a U(1) ghost number symmetry, which exhibits a mixed gravitational anomaly: On any Riemann surface, it is conserved up to a background charge proportional to the Euler characteristic. In [32], it was pointed out that the mixed gravitational anomaly, albeit consistent, cannot be canceled by the inflow of a relativistic classical action if the boundary $(1+1)d$ spin connection is to be matched with the bulk $(2+1)d$ spin connection. However, a non-relativistic inflow is possible using the renowned Wen-Zee topological term [33,34]. In this paper, we propose a relativistic inflow that matches the boundary $(1+1)d$ spin connection with a bulk SO(2) gauge field.

Another slightly bizarre feature of the mixed gravitational anomaly is the non-existence of an improved stress tensor with covariant anomalous conservation. Recall that a consistent anomaly requires the current to be defined via the variation of background fields, and the resulting anomalous conservation equations are generally not gauge-covariant. By adding Bardeen-Zumino currents [35], the consistent current can often be improved to a covariant one, *i.e.* with covariant anomalous conservation equations. The covariant currents are no longer equal to the variation of background fields, and do not satisfy the Wess-Zumino consistency condition. We show that this improvement is not possible for the mixed gravitational anomaly at hand.

The rest of this paper is organized as follows. Section 1.1 defines the consistency and locality conditions. Section 2 concerns pure anomalies, by first reviewing the anomaly descent and inflow of perturbative pure anomalies, and then examining the finite Wess-Zumino condition for their global versions. Section 3 explores the mixed U(1)-gravitational anomaly and its connection to the isotopy anomaly and periodicity anomaly of topological defect lines. Section 4 discusses an important subtlety regarding the anomaly of finite subgroups of U(1). Section 5 surveys the holomorphic $bc$ ghost system and finds it to realize every exotic consistent anomaly discussed in this paper. Section 6 ends with concluding remarks. Appendix A reviews the Bardeen-Zumino counter-terms for pure gravitational anomaly, and constructs its counterpart for the mixed anomaly. Appendix B proves that the consistent mixed gravitational anomaly does not have covariant counterpart.

## 1.1 Consistency and locality

't Hooft anomalies satisfy two conditions: (finite Wess-Zumino) consistency and locality. Consistency amounts to the commutativity of the diagram (1.2) up to $2\pi\mathbb{Z}$ phase differences.

**Condition 1.1** (Consistency). *For two arbitrary background diffeomorphism/gauge transformations $\Lambda_1$ and $\Lambda_2$, the anomalous phases satisfy*

$$\alpha[\Phi, \Lambda_2\Lambda_1] - \alpha[\Phi^{\Lambda_1}, \Lambda_2] - \alpha[\Phi, \Lambda_1] \in 2\pi\mathbb{Z}. \tag{1.3}$$

Locality amounts to the following two properties:

1. Under general background diffeomorphism/gauge transformations $\Lambda$, the anomalous phase $\alpha[\Phi, \Lambda]$ is a local functional of $\Phi$.

2. Under infinitesimal background diffeomorphism/gauge transformations $\Lambda$, the anomalous phase $\alpha[\Phi, \Lambda]$ is a local functional of $\Phi$ and $\Lambda$, and vanishes when $\Phi = 0$. For the gravitational background, $\Phi = 0$ means that the spin connection (or Levi-Civita connection) vanishes, with no further constraint on the vielbein.

An argument for the second locality property can be made as follows. For continuous symmetries, the divergence of the Noether current $J^\mu$ should vanish in correlation functions up to contact terms,

$$\langle \nabla_\mu J^\mu(x) \cdots \rangle \Big|_{\Phi=0} = \text{contact terms}. \tag{1.4}$$

Had the second locality property been false, this contact structure would be violated by the anomalous Ward identities. The first locality property can be viewed as an extension of the second locality property to large background diffeomorphism/gauge transformations. The two locality properties above can be stated in more precise terms by the following locality condition.

**Condition 1.2** (Locality). *Let $\mathscr{G}$ be the space of all background differeomorphism/gauge transformations, with connected components $\mathscr{G}_n$ for $n = 0, 1, 2, \cdots$, and with $\mathscr{G}_0$ containing the trivial transformation. The anomalous phase $\alpha[\Phi, \Lambda]$ takes the form*

$$\alpha[\Phi, \Lambda] = \sum_i \kappa_i(n)\, \mathcal{A}_i[\Phi, \Lambda] + \theta(n), \tag{1.5}$$

*where $\mathcal{A}_i[\Phi, \Lambda]$ is a basis of independent local functionals that vanish in the trivial background $\Phi = 0$, and $\theta(0) = 0$.*

# 2 Pure anomalies

This section first reviews the perturbative pure gravitational and U(1) anomalies in non-spin QFT, and then examines the finite Wess-Zumino (fWZ) consistency condition for global anomalies. We derive a weaker quantization condition on the anomaly coefficients than that of inflow. A comparison can be found in Table 1.

## 2.1 Perturbative pure anomalies

We begin by reviewing the well-known perturbative pure anomalies. Consider a $(1+1)d$ non-spin QFT with U(1) internal symmetry coupled to a background metric $g_{\mu\nu}$ and a background U(1) gauge field $A$. We parameterize the background metric by the zweibein $e_\mu^a$ and write $g_{\mu\nu} = e_\mu^a e_\nu^b \delta_{ab}$. We use $\mu, \nu, \dots$ to denote spacetime indices, and $a, b, \dots$ to denote frame indices. Under diffeomorphisms ($\xi$), local frame rotations ($\theta$), and U(1) gauge transformations ($\lambda$) the background zweibein and the background U(1) gauge field transform as

$$\delta e^a = -\theta^a{}_b e^b + \mathcal{L}_\xi e^a , \quad \delta A = d\lambda + \mathcal{L}_\xi A , \tag{2.1}$$

where $\mathcal{L}_\xi$ denotes the Lie-derivative.

The effective action $W[e, A] = -\log Z[e, A]$ is a complex-valued functional of the background fields, and is in general non-local. The infinitesimal part of the anomalous phase is a local functional linear in the gauge parameters,

$$\alpha[e, A, \theta, \xi, \lambda] = \mathcal{A}_\theta[e, A, \theta] + \mathcal{A}_\xi[e, A, \xi] + \mathcal{A}_\lambda[e, A, \lambda] . \tag{2.2}$$

The effective action shifts by

$$W[e + \delta e, A + \delta A] = W[e, A] - i\left(\mathcal{A}_\theta[e, A, \theta] + \mathcal{A}_\xi[e, A, \xi] + \mathcal{A}_\lambda[e, A, \lambda]\right) . \tag{2.3}$$

The anomalous phases $\mathcal{A}_\theta$, $\mathcal{A}_\xi$ and $\mathcal{A}_\lambda$ are constrained by the Wess-Zumino consistency condition [36]

$$\delta_{\chi_1} \mathcal{A}_{\chi_2} - \delta_{\chi_2} \mathcal{A}_{\chi_1} = \mathcal{A}_{[\chi_2, \chi_1]} , \quad \text{for} \quad \chi = \theta , \xi , \lambda . \tag{2.4}$$

### Descent equations

A large class of solutions to the Wess-Zumino consistency condition are obtained by the descent equations

$$\mathcal{I}^{(4)} = d\mathcal{I}^{(3)} , \quad \delta\mathcal{I}^{(3)} = d\mathcal{I}^{(2)} , \quad \mathcal{A} = 2\pi \int_{\mathcal{M}_2} \mathcal{I}^{(2)} , \tag{2.5}$$

where $\mathcal{I}^{(3)}$ and $\mathcal{I}^{(4)}$ are formal 3- and 4-forms. The 4-form anomaly polynomial responsible for the pure gravitational and U(1) anomalies is

$$\mathcal{I}^{(4)} = \frac{1}{(2\pi)^2} \left[ \frac{\kappa_{R^2}}{48} \text{tr} \, (R \wedge R) + \frac{\kappa_{F^2}}{2} F \wedge F \right], \tag{2.6}$$

where $R_{ab} = \frac{1}{2} e_a^\mu e_b^\nu R_{\mu\nu\rho\sigma} dx^\rho dx^\sigma$ and $F = dA$. The descent 3-form is

$$\mathcal{I}^{(3)} = \frac{1}{(2\pi)^2} \left[ \frac{\kappa_{R^2}}{48} \text{CS}(\omega) + \frac{\kappa_{F^2}}{2} A \wedge F \right], \tag{2.7}$$

and the anomalous phases are

$$\mathcal{A}_\theta = \frac{\kappa_{R^2}}{96\pi} \int_{\mathcal{M}_2} \theta^{ab} R_{ba} \,, \quad A_\xi = 0 \,, \quad \mathcal{A}_\lambda = \frac{\kappa_{F^2}}{4\pi} \int_{\mathcal{M}_2} \lambda \, F \,. \tag{2.8}$$

### Inflow mechanism

An anomaly that solves the descent equations has a natural bulk classical action. Consider

$$S_{\text{bulk}} = \frac{ik_{R^2}}{192\pi} \int_{\mathcal{M}_3} \text{CS}(\omega) + \frac{ik_{F^2}}{4\pi} \int_{\mathcal{M}_3} \text{CS}(A) \,, \tag{2.9}$$

which, to be well-defined, must have quantized levels[3]

$$\frac{k_{R^2}}{8}, \; k_{F^2} \in 2\mathbb{Z} \,. \tag{2.11}$$

If $\mathcal{M}_3$ is a three-manifold with boundary $\partial\mathcal{M}_3 = \mathcal{M}_2$, then the classical action on $\mathcal{M}_3$ contributes the following amount of anomaly to the $(1+1)d$ non-spin QFT on $\mathcal{M}_2$,

$$\Delta\kappa_{R^2} = -\frac{1}{2} k_{R^2} \,, \quad \Delta\kappa_{F^2} = -k_{F^2} \,. \tag{2.12}$$

For the coupled system to be free of anomalies, the quantization conditions (2.11) on the Chern-Simons levels $k_{R^2}$ and $k_{F^2}$ translate to

$$\frac{1}{8}\kappa_{R^2} \,, \; \frac{1}{2}\kappa_{F^2} \in \mathbb{Z} \,. \tag{2.13}$$

---

[3]On a closed manifold $\mathcal{M}_3$, the Chern-Simons action (2.9) is required to be invariant under background diffeomorphism and U(1) gauge transformations. One way to manifest the invariance property is to rewrite the action as

$$S = \frac{ik_{R^2}}{192\pi} \int_{\mathcal{M}_4} \text{tr} \, R \wedge R + \frac{ik_{F^2}}{4\pi} \int_{\mathcal{M}_4} F \wedge F \,, \tag{2.10}$$

where $\mathcal{M}_4$ is a four manifold such that $\partial\mathcal{M}_4 = \mathcal{M}_3$. For (2.10) to be independent of the choice of $\mathcal{M}_4$, the levels $k_{R^2}$ and $k_{F^2}$ must be quantized as in (2.11).

## Bardeen-Zumino counter-term

The Bardeen-Zumino counter-term provides a trade-off between the frame rotation anomaly and the diffeomorphism anomaly [37]. The conventional choice eliminates the former in favor of the latter. The counter-term is constructed from the zwiebein $e^a{}_\mu$, with the explicit form given in (A.1). The modified effective action is

$$W'[e, A] \equiv W[e, A] + S_{\mathrm{BZ}}[e] \,, \tag{2.14}$$

such that under local frame rotations,

$$\delta_\theta S_{\mathrm{BZ}} = i\mathcal{A}_\theta \,. \tag{2.15}$$

Hence, the new effective action $W'[e, A]$ transforms as

$$W'[e + \delta e, A + \delta A] = W'[e, A] - i \left( \mathcal{A}_\lambda[e, A, \lambda] + \mathcal{A}'_\xi[e, A, \xi] \right) \,, \tag{2.16}$$

with a nonzero anomalous phase $\mathcal{A}'_\xi[e, A, \xi]$ under diffeomorphism,

$$\mathcal{A}'_\xi = i\delta_\xi S_{\mathrm{BZ}} = \frac{\kappa_{R^2}}{96\pi} \int_{\mathcal{M}_2} \partial_\mu \xi^\nu d\Gamma^\mu{}_\nu \,. \tag{2.17}$$

## Anomalous conservation and covariant improvement

The anomalous phases (2.8) imply the anomalous conservation equations

$$\langle \nabla_\mu T^{\mu\nu}(x) \rangle = -\frac{2\pi i}{\sqrt{g}} \frac{\delta \mathcal{A}'_\xi[e, A, \xi]}{\delta \xi_\nu} = \frac{i\kappa_{R^2}}{48} \frac{1}{\sqrt{g}} g^{\nu\lambda} \partial_\mu \left( \sqrt{g} \varepsilon^{\rho\sigma} \partial_\rho \Gamma^\mu{}_{\lambda\sigma} \right) \,,$$
$$\langle \nabla^\mu J_\mu(x) \rangle = -\frac{2\pi}{\sqrt{g}} \frac{\delta \mathcal{A}_\lambda[e, A, \lambda]}{\delta \lambda(x)} = -\frac{\kappa_{F^2}}{4} \varepsilon^{\mu\nu} F_{\mu\nu} \,. \tag{2.18}$$

Note that the first equation is not covariant. In technical terms, these are consistent anomalies and not covariant anomalies [35]. To arrive at the latter, the stress tensor $T^{\mu\nu}$ must be improved by

$$\mathcal{T}^{\mu\nu} = T^{\mu\nu} - \frac{i\kappa_{R^2}}{48} \nabla_\lambda \left( \Gamma^{(\underline{\mu}\lambda}{}_\sigma \varepsilon^{\underline{\nu})\sigma} - \Gamma^{\lambda(\mu}{}_\sigma \varepsilon^{\nu)\sigma} - \Gamma^{(\mu\nu)}{}_\sigma \varepsilon^{\lambda\sigma} \right) \,, \tag{2.19}$$

The anomalous conservation equation for the improved stress tensor $\mathcal{T}^{\mu\nu}$ takes the covariant form

$$\langle \nabla_\mu \mathcal{T}^{\mu\nu}(x) \rangle = \frac{i\kappa_{R^2}}{48} \nabla_\mu (R^{\mu\nu}{}_{\rho\sigma} \varepsilon^{\rho\sigma}) \,. \tag{2.20}$$

## Operator product in CFT

On flat space, the two-point functions of the stress tensor $T_{\mu\nu}$ and the conserved current $J_\mu$ are constrained by conformal symmetry to be

$$\langle T_{zz}(z,\bar{z})T_{zz}(0)\rangle = \frac{c}{2z^4}, \quad \langle T_{\bar{z}\bar{z}}(z,\bar{z})T_{\bar{z}\bar{z}}(0)\rangle = \frac{\bar{c}}{2\bar{z}^4}, \tag{2.21}$$

and

$$\langle J_z(z,\bar{z})J_z(0)\rangle = \frac{k}{z^2}, \quad \langle J_{\bar{z}}(z,\bar{z})J_{\bar{z}}(0)\rangle = \frac{\bar{k}}{\bar{z}^2}. \tag{2.22}$$

The remaining components

$$\langle T_{zz}(z,\bar{z})T_{z\bar{z}}(0)\rangle, \ \langle T_{\bar{z}\bar{z}}(z,\bar{z})T_{z\bar{z}}(0)\rangle, \ \langle T_{zz}(z,\bar{z})T_{\bar{z}\bar{z}}(0)\rangle, \ \langle T_{z\bar{z}}(z,\bar{z})T_{z\bar{z}}(0)\rangle, \ \langle J_z(z,\bar{z})J_{\bar{z}}(0)\rangle$$

are contact terms with coefficients related to the anomalies. The above two point functions can be obtained from the Ward identities implied by the anomalous conservation equations (3.14), giving

$$c_- \equiv c - \bar{c} = \kappa_{R^2}, \quad k_- \equiv k - \bar{k} = \kappa_{F^2}. \tag{2.23}$$

The discussion of the $\langle TJ\rangle$ two-point functions is deferred to Section 3.1.

## 2.2 Global gravitational anomaly

Let us now examine the pure anomaly of large diffeomorphisms.[4] Since we do not assume time-reversal symmetry, orientation-reversing operations such as reflections are excluded. For concreteness, consider a $(1+1)d$ non-spin CFT on a torus with complex moduli $\tau$ and a flat metric

$$ds^2 = |dx^1 + \tau dx^2|^2, \quad x^\mu \cong x^\mu + 2\pi\mathbb{Z}. \tag{2.24}$$

The orientation-preserving large diffeomorphisms that respect the periodicity of the coordinates $x^\mu$ are

$$\begin{pmatrix} x^1 \\ x^2 \end{pmatrix} \rightarrow \begin{pmatrix} x'^1 \\ x'^2 \end{pmatrix} = \begin{pmatrix} a & -b \\ -c & d \end{pmatrix}\begin{pmatrix} x^1 \\ x^2 \end{pmatrix} \quad \text{for} \quad ad - bc = 1 \quad \text{and} \quad a, b, c, d \in \mathbb{Z}, \tag{2.25}$$

and form the mapping class group $SL(2,\mathbb{Z})$. It is generated by

$$S = \begin{pmatrix} 0 & 1 \\ -1 & 0 \end{pmatrix}, \quad T = \begin{pmatrix} 1 & -1 \\ 0 & 1 \end{pmatrix}, \tag{2.26}$$

---

[4]Essentially the same analysis as this subsection was done in [38] and generalized to arbitrary genera, using the language of conformal field theory as analytic geometric on the universal moduli space of Riemann surfaces [39].

which satisfy the relations

$$S^4 = 1, \quad (ST)^3 = S^2 \,. \tag{2.27}$$

The form of the metric (2.24) is preserved, modulo Weyl transformations, by $\mathrm{SL}(2, \mathbb{Z})$, with the complex moduli $\tau$ and the complex coordinate $w = x^1 + \tau x^2$ transformed as

$$\tau \to \tau' = \frac{a\tau + b}{c\tau + d}\,, \quad w \to w' = x'^1 + \tau' x'^2 = \frac{w}{c\tau + d}\,. \tag{2.28}$$

### Torus partition function

Suppose the partition function on a flat torus does not vanish identically over all moduli.[5] Under $\mathrm{SL}(2, \mathbb{Z})$, the only possible dependence on the flat background geometry that is compatible with locality is through the volume integral $\int d^2x \sqrt{g}$. However, an anomalous phase proportional to the volume violates the fWZ consistency condition 1.1, with $\Lambda_1$ an $\mathrm{SL}(2, \mathbb{Z})$ transformation, and $\Lambda_2$ a Weyl transformation. Hence, the torus partition function must be invariant under $\mathrm{SL}(2, \mathbb{Z})$ up to $\tau$-independent anomalous phases[6]

$$Z\left(\frac{a\tau + b}{c\tau + d}, \frac{a\bar{\tau} + b}{c\bar{\tau} + d}\right) = Z(\tau, \bar{\tau})\, e^{i\theta(a,b,c,d)} \,. \tag{2.30}$$

By the fWZ consistency condition 1.1, the general phases $\theta(a, b, c, d)$ are determined from the phases $\theta_S$ and $\theta_T$ of the $S$ and $T$ generators, *i.e.*

$$Z\left(-\frac{1}{\tau}, -\frac{1}{\bar{\tau}}\right) = Z(\tau, \bar{\tau})\, e^{i\theta_S} \,, \quad Z(\tau + 1, \bar{\tau} + 1) = Z(\tau, \bar{\tau})\, e^{i\theta_T} \,. \tag{2.31}$$

The chiral central charge is related to the $T$ anomalous phase by $2\pi c_- = -24\,\theta_T$. Under the relations (2.27), fWZ constrains[7]

$$2\theta_S \in 2\pi\mathbb{Z}\,, \quad \theta_S + 3\theta_T \in 2\pi\mathbb{Z}\,. \tag{2.33}$$

---

[5]The usual reason for a partition function to vanish identically is the existence of anti-commuting zero modes.

[6]On the flat torus (in Cartesian coordinates (2.24)) where the Christoffel symbols all vanish, no local integral term can contribute. Therefore, the fWZ consistency condition 1.1 modulo phase redefinitions defines the first group cohomology with $\mathrm{U}(1)$ coefficients. This subsection is essentially an exercise computing

$$H^1(P\mathrm{SL}(2, \mathbb{Z}), \mathrm{U}(1)) = \mathbb{Z}_6\,, \quad H^1(\mathrm{SL}(2, \mathbb{Z}), \mathrm{U}(1)) = \mathbb{Z}_{12}\,. \tag{2.29}$$

[7]Note that the anomalous phases form a representations of $P\mathrm{SL}(2, \mathbb{Z})$, defined by the relations

$$S^2 = (ST)^3 = 1\,. \tag{2.32}$$

This is physically expected because $S^2$ is charge conjugation, and acts trivially on a torus with no operator insertions.

There are two scenarios:

$$
\begin{array}{lll}
\text{(i)} \quad \theta_S,\, 3\theta_T \in 2\pi\mathbb{Z} & \Rightarrow\ Z(\tau,\bar\tau) = Z(-1/\tau, -1/\bar\tau)\,, & c_- \in 8\mathbb{Z}\,, \\[2mm]
\text{(ii)} \quad \theta_S,\, 3\theta_T \in 2\pi\left(\mathbb{Z} + \dfrac{1}{2}\right) & \Rightarrow\ Z(\tau,\bar\tau) = -Z(-1/\tau, -1/\bar\tau)\,, & c_- \in 8\mathbb{Z} + 4\,.
\end{array}
\tag{2.34}
$$

In scenario (ii), $Z(\tau = i,\ \bar\tau = -i)$ on the square torus must either blow up or vanish. The former means that the spectrum exhibits Hagedorn growth, which violates our expectation of QFT in finite volume.[8] The latter violates reflection-positivity. See Figure 1. Hence, a reflection-positive CFT must fall into scenario (i).[9]

More generally, an $ST^n$ transformation produces a phase factor

$$
e^{i(\theta_S + n\theta_T)} =
\begin{cases}
1 & c_- \in 24\mathbb{Z}\,, \\
\omega^{\pm n} & c_- \in 24\mathbb{Z} \pm 8\,, \\
-(-)^n & c_- \in 24\mathbb{Z} + 12\,, \\
-(-\omega)^{\pm n} & c_- \in 24\mathbb{Z} \pm 4\,,
\end{cases}
\tag{2.35}
$$

where $\omega = e^{\frac{2}{3}\pi i}$. An immediate consequence is that the partition function $Z(\tau,\bar\tau)$ must vanish at the $S$-invariant point $\tau = i$ and/or the $ST$-invariant point $\tau = \omega$ whenever $c_- \notin 24\mathbb{Z}$. More specifically, the vanishing points in the standard fundamental domain are

$$
\tau =
\begin{cases}
\omega & c_- \in 24\mathbb{Z} \pm 8\,, \\
i & c_- \in 24\mathbb{Z} + 12\,. \\
i,\, \omega & c_- \in 24\mathbb{Z} \pm 4\,.
\end{cases}
\tag{2.36}
$$

As a check, the chiral half of the $(E_8)_1$ WZW model has $c_- = 8$, and its torus partition function $Z(\tau) = J(\tau)^{\frac{1}{3}}$ indeed vanishes at $\tau = \omega$.

### Torus one-point function

One can derive similar conditions by looking at the torus one-point function

$$
G(\tau,\bar\tau) = \langle \mathcal{O}_{h,\bar h}(w,\bar w)\rangle_{T^2_\tau}\,.
\tag{2.37}
$$

of a local operator $\mathcal{O}_{h,\bar h}$ that has definite holomorphic and anti-holomorphic weights $h$ and $\bar h$ but is not required to be a primary. By translational invariance, the torus one-point function

---

[8]See [40] for a discussion. In the following we always assume that the torus partition function (for non-compact CFTs normalized by the volume) does not blow up.

[9]Many non-reflection-positive CFTs such as the $c < 0$ minimal models still have positive torus partition functions. They must also fall into scenario (i).

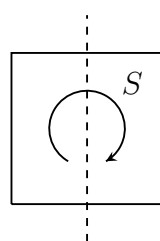

Figure 1: The square torus $\tau = i$, $\bar{\tau} = -1$ is symmetric under 90 degree rotations (modular $S$) and reflections. The partition function on the square torus transforms with a phase $\theta_S$ under the former, and must be positive in a reflection-positive theory due to the later.

does not depend on the coordinate $w$ of the operator insertion. Under $S$, it transforms as

$$\langle \mathcal{O}_{h,\bar{h}}(w,\bar{w})\rangle_{T^2_\tau} = e^{-i\theta_S}\langle \mathcal{O}_{h,\bar{h}}(w,\bar{w})\rangle_{T^2_{-1/\tau}} = e^{-i\theta_S}\tau^{-h}\bar{\tau}^{-\tilde{h}}\langle \mathcal{O}'_{h,\bar{h}}(w/\tau,\bar{w}/\bar{\tau})\rangle_{T^2_{-1/\tau}}. \quad (2.38)$$

Under $T$, there is no conformal factor. In summary,

$$G\left(-\frac{1}{\tau}, -\frac{1}{\bar{\tau}}\right) = e^{i\theta_S}\tau^h\bar{\tau}^{\bar{h}}\,G(\tau,\bar{\tau})\,, \quad G(\tau+1,\bar{\tau}+1) = e^{i\theta_T}\,G(\tau,\bar{\tau})\,. \quad (2.39)$$

The anomalous phases $\theta_S$ and $\theta_T$ satisfy the quantization conditions

$$2\theta_S \in 2\pi\left(\mathbb{Z} + \frac{\ell}{2}\right)\,, \quad \theta_S + 3\theta_T \in 2\pi\mathbb{Z}\,, \quad (2.40)$$

where $\ell = h - \bar{h}$ is the spin of the operator $\mathcal{O}$.

In a given theory, the torus one-point functions for different operators can have different $\theta_S$, but they must have the same $\theta_T$, which is related to the chiral central charge by $2\pi c_- = -24\,\theta_T$. A CFT with a non-vanishing torus one-point function of an operator operator $\mathcal{O}_{h,\bar{h}}$ of odd spin necessary contains anti-commuting fields, *i.e.* ghosts if the QFT is non-spin. This is because $\mathcal{O}_{h,\bar{h}}$ must appear in the OPE of some real operator $\mathcal{O}_{h',\bar{h}'}$ with itself,

$$\mathcal{O}_{h',\bar{h}'}(z_1,\bar{z}_1)\mathcal{O}_{h',\bar{h}'}(z_2,\bar{z}_2) \;\ni\; C(z_1-z_2)^{h-2h'}(\bar{z}_1-\bar{z}_2)^{\bar{h}-2\bar{h}'}\mathcal{O}_{h,\bar{h}}\left(\frac{z_1+z_2}{2}, \frac{\bar{z}_1+\bar{z}_2}{2}\right)\,. \quad (2.41)$$

Exchanging $z_1$ and $z_2$ produces a sign since

$$(-)^{h-\bar{h}-2(h'-\bar{h}')} = -1\,. \quad (2.42)$$

For the OPE coefficient $C$ to be non-zero, the operator $\mathcal{O}_{h',\bar{h}'}$ must therefore be anti-commuting (Grassmann-valued) to produce a compensating sign.

- If the torus one-point function for at least one operator of even spin does not vanish identically over all torus moduli, then we recover the previous condition (2.34), hence $c_- \in 4\mathbb{Z}$.

- If the torus one-point functions for at least one operator of odd spin does not vanish identically over all torus moduli — which can only happen in the presence of anti-commuting fields, *i.e.* ghosts if the CFT is non-spin — then (2.40) leads to $c_- \in 4\mathbb{Z}+2$.

### Quantization of the chiral central charge

The preceding results can be summarized as follows.

**Lesson 2.1.** *The chiral central charge of a non-spin CFT satisfies $c_- \in 2\mathbb{Z}$ if at least one torus one-point function does not vanish identically over all moduli of the torus. If the torus partition function itself does not vanish identically, then $c_- \in 4\mathbb{Z}$. If the partition function is positive on the square torus (true if reflection-positive), then $c_- \in 8\mathbb{Z}$.*[10]

Note that a $(2+1)d$ bulk gravitational Chern-Simons action can cancel the global gravitational anomaly if $c_- \in 8\mathbb{Z}$, which is guaranteed for reflection-positive CFTs. If not reflection-positive and $c_- \notin 8\mathbb{Z}$, then the global gravitational anomaly is consistent but more exotic. The holomorphic $bc$ ghost system realizes $c_- \in -2 + 24\mathbb{Z}$.

## 2.3  Global U(1) anomaly

Consider a $(1+1)d$ non-spin QFT with U(1) global symmetry on a genus-$g$ Riemann surface $\Sigma$. Let $\mathcal{C}_i$ for $i = 1, \cdots, 2g$ be a basis of non-contractable cycles on the Riemann surface $\Sigma$, with intersection matrix $\Omega$. The winding numbers of the gauge transformation $\lambda$ are

$$\vec{m}[\lambda] = \frac{1}{2\pi} \int_{\vec{\mathcal{C}}} d\lambda \,. \tag{2.43}$$

The locality condition 1.2 dictates that the anomalous phase takes the form

$$\alpha[A, \lambda] = -\frac{\kappa(\vec{m}[\lambda])}{4\pi} \int_\Sigma d\lambda\, A + \sum_i \frac{\kappa_i'(\vec{m}[\lambda])}{2\pi} \int_\Sigma f_i(\lambda)\, F + \theta(\vec{m}[\lambda]) \,, \tag{2.44}$$

where $f_i$ is a basis of periodic functions,

$$f_i(\lambda + 2\pi) = f_i(\lambda) \,, \tag{2.45}$$

and $\kappa$, $\kappa_i'$, $\theta$ are functions that satisfy

$$\kappa(0) = \kappa_{F^2} \,, \quad \theta(0) = 0 \,. \tag{2.46}$$

---

[10]The condition $c_- \in 2\mathbb{Z}$ was also found in the classification of $(2+1)d$ non-spin invertible topological orders by BF categories [41]. There is no known non-spin invertible topological order that realizes the minimal chiral central charge $c_- = \pm 2$. We thank Xiao-Gang Wen for pointing this out to us.

Let us focus on background gauge orbits that are flat, so that $\kappa'_i$ does not appear. Consider two large background gauge transformations $\lambda_1$ and $\lambda_2$ with nontrivial windings. For shorthand, we write

$$\vec{m}_1 \equiv \vec{m}[\lambda_1], \quad \vec{m}_2 \equiv \vec{m}[\lambda_2], \quad \vec{m}_{12} \equiv \vec{m}[\lambda_1 + \lambda_2] = \vec{m}_1 + \vec{m}_2. \tag{2.47}$$

The fWZ consistency condition 1.1 requires that

$$\left[ -\frac{\kappa(\vec{m}_{12})}{4\pi} \int_\Sigma d(\lambda_1 + \lambda_2) \, A + \theta(\vec{m}_{12}) \right] - \left[ \frac{\kappa(\vec{m}_2)}{4\pi} \int_\Sigma d\lambda_2 (A + d\lambda_2) + \theta(\vec{m}_2) \right]$$
$$- \left[ \frac{\kappa(\vec{m}_1)}{4\pi} \int_\Sigma d\lambda_1 \, A + \theta(\vec{m}_1) \right] \equiv 0 \mod 2\pi. \tag{2.48}$$

The above can be reorganized into

$$\left[ -\pi\kappa(\vec{m}_2) \, \vec{m}_1 \cdot \Omega \cdot \vec{m}_2 + \theta(\vec{m}_{12}) - \theta(\vec{m}_1) - \theta(\vec{m}_2) \right] - \left[ \frac{\kappa(\vec{m}_{12}) - \kappa(\vec{m}_1)}{4\pi} \int_\Sigma d\lambda_1 \, A \right]$$
$$- \left[ \frac{\kappa(\vec{m}_{12}) - \kappa(\vec{m}_2)}{4\pi} \int_\Sigma d\lambda_2 \, A \right] \equiv 0 \mod 2\pi, \tag{2.49}$$

where we used

$$\frac{1}{4\pi^2} \int_\Sigma d\lambda_1 d\lambda_2 = \vec{m}_1 \cdot \Omega \cdot \vec{m}_2. \tag{2.50}$$

Because $A$ is an arbitrary flat connection and $\lambda_1$, $\lambda_2$ are independent and arbitrary, the coefficients in second and third brackets must separately vanish. Hence,

$$\kappa(\vec{m}[\lambda]) = \kappa(0) = \kappa_{F^2} \tag{2.51}$$

is a constant.

We left with

$$- \pi\kappa_{F^2} \, \vec{m}_1 \cdot \Omega \cdot \vec{m}_2 + \theta(\vec{m}_{12}) - \theta(\vec{m}_1) - \theta(\vec{m}_2) \equiv 0 \mod 2\pi, \tag{2.52}$$

For concreteness, let $\Sigma$ be a torus, and choose a basis of cycles $\mathcal{C}_i$ with intersection matrix

$$\Omega = \begin{pmatrix} 0 & 1 \\ -1 & 0 \end{pmatrix}. \tag{2.53}$$

With

$$\vec{m}_1 = (1,0) \quad \vec{m}_2 = (-1,0), \tag{2.54}$$

and separately

$$\vec{m}_1 = (0,1), \quad \vec{m}_2 = (0,-1), \tag{2.55}$$

together with (2.46), we find

$$\theta(1,0) + \theta(-1,0) \equiv \theta(0,1) + \theta(0,-1) \equiv 0 \mod 2\pi \,. \tag{2.56}$$

With

$$\vec{m}_1 = (m-1, n)\,, \quad \vec{m}_2 = (1,0)\,, \tag{2.57}$$

and separately

$$\vec{m}_1 = (m, n-1)\,, \quad \vec{m}_2 = (0,1)\,, \tag{2.58}$$

we find recurrence relations on $\theta(m,n)$ for $(m,n)$ in the first quadrant,

$$\begin{aligned}
\theta(m,n) &\equiv \theta(m-1,n) + \theta(1,0) - \pi\kappa_{F^2}n \mod 2\pi\,, \\
\theta(m,n) &\equiv \theta(m,n-1) + \theta(0,1) + \pi\kappa_{F^2}m \mod 2\pi\,.
\end{aligned} \tag{2.59}$$

Similarly, there are recurrence relations for $(m,n)$ in the three other quadrants. The solution in all quadrants is

$$\theta(m,n) = \theta(1,0)m + \theta(0,1)n - \pi\kappa_{F^2}mn \,. \tag{2.60}$$

Plugging this solution back into (2.52), we find the quantization condition

$$\kappa_{F^2} \in \mathbb{Z} \,. \tag{2.61}$$

The quantization condition (2.61) is weaker than the quantization condition (2.13) expected from the inflow of $(2+1)d$ bulk U(1) Chern-Simons.

## Mixing with the modular transforms

Let $P_{m,n}$ denote a background U(1) gauge transformation with winding numbers $(m,n)$. From the fWZ consistency condition 1.1 for the relations

$$P_{1,0}\,S = S\,P_{0,1}\,, \quad T\,P_{1,0} = P_{1,1}\,T \,, \tag{2.62}$$

one deduces

$$\theta(1,0) = \theta(0,1) = \pi\kappa_{F^2} \,. \tag{2.63}$$

## Quantization of the level

In CFT, the anomaly coefficient and the level are related by $\kappa_{F^2} = k_-$.

**Lesson 2.2.** *The level $k_-$ of a U(1) current algebra in a non-spin CFT must satisfy $k_- \in \mathbb{Z}$ if the flavored torus partition function does not vanish identically over all moduli of the torus and all flat gauge backgrounds.*

Note that a $(2+1)d$ bulk U(1) Chern-Simons action can cancel the anomaly if $k_-$ is even. The holomorphic $bc$ ghost system realizes $k_- = 1$.

# 3  Mixed U(1)-gravitational anomaly

This section examines the mixed U(1)-gravitational anomaly a $(1+1)d$ non-spin QFT. In the first part of this section, we characterize the mixed gravitational anomaly by descent and inflow, examine the possibility of a covariant improvement, and study the imprint of the anomaly on local operator products in CFT. In the second part, we study the mixed gravitational anomaly from the perspective of topological defects, and show that the mixed gravitational anomaly gives rise to an isotopy anomaly.

## 3.1  Perturbative mixed U(1)-gravitational anomaly

In the following, $A$ and $F$ denote the U(1) connection and field strength, and $\omega$ denotes the spin connection, with $R$ its field strength. We use $a, b, \dots$ to denote frame indices.

### Descent equations

The mixed gravitational anomaly is described by the anomaly polynomial,

$$\mathcal{I}^{(4)} = \frac{\kappa_{FR}}{(2\pi)^2} F \wedge \left( \varepsilon^{ab} R_{ba} \right) . \tag{3.1}$$

The descent 3-form is[11]

$$\mathcal{I}^{(3)} = \frac{1}{(2\pi)^2} \left[ \frac{\kappa_{FR}}{2} A \wedge \varepsilon^{ab} R_{ba} + \frac{\kappa_{FR}}{2} F \wedge \varepsilon^{ab} \omega_{ba} + sd \left( A \wedge \varepsilon^{ab} \omega_{ba} \right) \right] , \tag{3.2}$$

where the ambiguity $s$ is related to the freedom of adding the Bardeen counter-term

$$S_{\mathrm{B}} = -\frac{is'}{2\pi} \int A \wedge \left( \varepsilon^{ab} \omega_{ba} \right) . \tag{3.3}$$

Its addition to the action shifts the ambiguity $s$ to $s + s'$. The anomalous phases are

$$\mathcal{A}_\lambda = \frac{1}{2\pi} \left( \frac{\kappa_{FR}}{2} - s \right) \int_{\mathcal{M}_2} \lambda \varepsilon^{ab} R_{ba} , \quad \mathcal{A}_\theta = \frac{1}{2\pi} \left( \frac{\kappa_{FR}}{2} + s \right) \int_{\mathcal{M}_2} \theta^{ab} \varepsilon_{ba} F , \quad \mathcal{A}_\xi = 0 . \tag{3.4}$$

### Inflow mechanism

Can the mixed gravitational anomaly of a $(1+1)d$ non-spin QFT be canceled by coupling to a $(2+1)d$ classical action? When the $(2+1)d$ spacetime is a product manifold $\mathcal{M}_3 = \mathcal{M}_2 \times$

---

[11]The descent equation $\mathcal{I}^{(4)} = d\mathcal{I}^{(3)}$ is insensitive to the addition of exact terms (total derivatives).

$[0, \infty)$, one could consider the $(2+1)d$ classical action of the renowned Wen-Zee topological term [33, 34] relevant for the Hall viscosity in non-relativistic quantum Hall systems (see [42–44] for the connection)

$$\frac{ik_{FR}}{16\pi} \int_{\mathcal{M}_2 \times [0,\infty)} \varepsilon^{ab} \omega_{ab} \wedge F \,, \tag{3.5}$$

where $\mathcal{M}_2$ is the spatial manifold, and the anomaly coefficient $k_{FR}$ is also called the spin vector.[12] The above inflow action explicitly breaks $(2+1)d$ Lorentz invariance, and thus requires non-relativistic geometry to generalize to non-product manifolds.

We propose a slightly different inflow mechanism that preserves $(2+1)d$ Lorentz invariance. Consider the mixed Chern-Simons term

$$\frac{ik_{FR}}{4\pi} \int_{\mathcal{M}_3} A \wedge F_R \,, \tag{3.6}$$

where $\mathcal{M}_3$ is a three-dimensional manifold whose boundary is $\mathcal{M}_2$, and $F_R = dA_R$ is the field strength of a background SO(2) gauge field on $\mathcal{M}_3$. The matching condition at $\partial \mathcal{M}_3 = \mathcal{M}_2$ is such that the normal component of $A_R$ vanishes, and the tangent components of $A_R$ are identified with the boundary $(1+1)d$ spin connection by

$$A_R\big|_{\mathcal{M}_2} = \frac{1}{\zeta} \varepsilon^{ab} \omega_{ba} \,, \tag{3.7}$$

with a proportionality constant $\zeta$ to be fixed by flux quantization. The flux of $\varepsilon^{ab} \omega_{ba}$ can be computed as

$$\int_{\mathcal{M}_2} \varepsilon^{ab} R_{ba} = - \int_{\mathcal{M}_2} d^2x \sqrt{g} R = -4\pi\chi \,. \tag{3.8}$$

Depending on whether the theory is defined only on orientable Riemann surfaces, for instance when there is no time-reversal symmetry, or on general Riemann surfaces, the Euler characteristic is quantized as $\chi \in 2\mathbb{Z}$ or $\chi \in \mathbb{Z}$, respectively. Hence, flux quantization determines

$$\zeta = \begin{cases} 4 & \mathcal{M}_2 \text{ orientable} \,, \\ 2 & \mathcal{M}_2 \text{ general} \,. \end{cases} \tag{3.9}$$

To cancel the mixed gravitational anomaly of the $(1+1)d$ QFT, the Chern-Simons level is chosen to be

$$k_{FR} = 2\zeta \kappa_{FR} \,. \tag{3.10}$$

The quantization condition for the Chern-Simons level is

$$k_{FR} \in 2\mathbb{Z} \tag{3.11}$$

---

[12] We thank Xiao-Gang Wen and Juven Wang for bringing our attention to [33] and [34].

translates to a quantization condition on the mixed gravitational anomaly coefficient

$$\kappa_{FR} \in \begin{cases} \frac{1}{4}\mathbb{Z} & \mathcal{M}_2 \text{ orientable}, \\ \frac{1}{2}\mathbb{Z} & \mathcal{M}_2 \text{ general}. \end{cases} \tag{3.12}$$

We will see in Section 5 that the holomorphic $bc$ ghost system realizes $\kappa_{FR} \in \frac{1}{4} + \frac{1}{2}\mathbb{Z}$. It is in principle possible to derive a quantization condition on $\kappa_{FR}$ from fWZ alone without the need of inflow. However, to probe $\kappa_{FR}$ requires considering curved Riemann surfaces and is beyond the scope of this paper.

### Bardeen-Zumino counter-terms

By adding a mixed Bardeen-Zumino counter-term $S_{\text{BZ}}^{\text{mixed}}$ which we construct in Appendix A.2, the anomalous phase $\mathcal{A}_\theta$ under frame rotations can be completely canceled, while generating an extra contribution to the anomalous phase $A'_\xi$ under diffeomorphisms. In summary, the new anomalous phases are

$$\begin{aligned} \mathcal{A}'_\theta &= \mathcal{A}_\theta + i\delta_\theta S_{\text{BZ}}^{\text{mixed}} = 0, \\ \mathcal{A}'_\xi &= i\delta_\xi S_{\text{BZ}}^{\text{mixed}} = \frac{1}{2\pi}\left(\frac{\kappa_{FR}}{2} + s\right) \int_{\mathcal{M}_2} \partial_\mu \xi^\nu d(\varepsilon^\mu{}_\nu A), \\ \mathcal{A}'_\lambda &= \mathcal{A}_\lambda + i\delta_\lambda S_{\text{BZ}}^{\text{mixed}} = \frac{1}{2\pi}\int_{\mathcal{M}_2} \lambda\left[\kappa_{FR}\varepsilon^{ab}R_{ba} - \left(\frac{\kappa_{FR}}{2} + s\right)d(\varepsilon^\nu{}_\mu \Gamma^\mu{}_\nu)\right]. \end{aligned} \tag{3.13}$$

### Anomalous conservation and covariant improvement

The anomalous phases (3.13) give the non-covariant anomalous conservation equations for the consistent currents,

$$\begin{aligned} \langle \nabla_\mu T^{\mu\nu}(x) \rangle &= -\frac{2\pi i}{\sqrt{g}} \frac{\delta \mathcal{A}'_\xi[e, A, \xi]}{\delta \xi_\nu} = i\left(\frac{\kappa_{FR}}{2} + s\right)\frac{1}{\sqrt{g}} g^{\nu\lambda}\partial_\mu\left[\sqrt{g}\varepsilon^{\rho\sigma}\partial_\rho(\varepsilon^\mu{}_\lambda A_\sigma)\right], \\ \langle \nabla^\mu J_\mu(x) \rangle &= -\frac{2\pi}{\sqrt{g}} \frac{\delta \mathcal{A}'_\lambda[e, A, \lambda]}{\delta \lambda(x)} = \kappa_{FR}R + \left(\frac{\kappa_{FR}}{2} + s\right)\nabla_\rho(\varepsilon^{\rho\sigma}\varepsilon^\nu{}_\mu \Gamma^\mu{}_{\nu\sigma}). \end{aligned} \tag{3.14}$$

The conservation of U(1) can be covariantized by improving the consistent current $J^\mu$ with Bardeen-Zumino currents, which are terms that depend only on the background fields and vanish when $A_\mu = 0$ and $g_{\mu\nu} = \delta_{\mu\nu}$. More precisely, the improved current is

$$\mathcal{J}^\mu = J^\mu - (\frac{\kappa_{FR}}{2} + s)\varepsilon^{\mu\nu}\varepsilon^\rho{}_\sigma \Gamma^\sigma{}_{\rho\nu}, \tag{3.15}$$

which has a covariant form of the anomalous conservation equation

$$\langle \nabla^\mu \mathcal{J}_\mu(x) \rangle = -\frac{\kappa_{F^2}}{4}\varepsilon^{\mu\nu}F_{\mu\nu} + \kappa_{FR}R. \tag{3.16}$$

However, by an explicit computation in Appendix B, we show that no covariant improvement of the stress tensor exists.

## Operator product in CFT

In flat space CFT, the two-point functions between the U(1) current $J_\mu$ and the stress tensor $T_{\mu\nu}$ must take the form

$$\langle T_{zz}(z)J_z(0)\rangle = \frac{\alpha}{z^3}, \quad \langle T_{\bar{z}\bar{z}}(\bar{z})J_{\bar{z}}(0)\rangle = \frac{\bar{\alpha}}{\bar{z}^3}, \tag{3.17}$$

which imply the commutation relations[13]

$$[L_m, J_n] = -nJ_{m+n} + \frac{m(m+1)}{2}\alpha\delta_{m+n}, \quad [\bar{L}_m, \bar{J}_n] = -n\bar{J}_{m+n} + \frac{m(m+1)}{2}\bar{\alpha}\delta_{m+n}. \tag{3.18}$$

One also has the contact terms

$$\langle T_{zz}(z,\bar{z})J_{\bar{z}}(0)\rangle = 2\pi\beta\partial\delta^{(2)}(z,\bar{z}), \quad \langle T_{\bar{z}\bar{z}}(z,\bar{z})J_z(0)\rangle = 2\pi\bar{\beta}\bar{\partial}\delta^{(2)}(z,\bar{z}),$$
$$\langle T_{z\bar{z}}(z,\bar{z})J_{\bar{z}}(0)\rangle = 2\pi\gamma\bar{\partial}\delta^{(2)}(z,\bar{z}), \quad \langle T_{z\bar{z}}(z,\bar{z})J_z(0)\rangle = 2\pi\bar{\gamma}\partial\delta^{(2)}(z,\bar{z}). \tag{3.19}$$

Matching the above with the anomalous Ward identities implied by the anomalous conservation equations (3.14), we arrive at the relations

$$\alpha + 2\beta = 4(\frac{\kappa_{FR}}{2} - s), \quad \bar{\alpha} + 2\bar{\beta} = 4(\frac{\kappa_{FR}}{2} - s), \quad \gamma + \bar{\gamma} = -2(\frac{\kappa_{FR}}{2} - s),$$
$$\beta + \gamma = -(\frac{\kappa_{FR}}{2} + s), \quad \bar{\beta} + \bar{\gamma} = -(\frac{\kappa_{FR}}{2} + s),$$
$$\alpha + 2\bar{\gamma} = 2(\frac{\kappa_{FR}}{2} + s), \quad \bar{\alpha} + 2\gamma = 2(\frac{\kappa_{FR}}{2} + s). \tag{3.20}$$

In particular,

$$\alpha + \bar{\alpha} = 4\kappa_{FR} \tag{3.21}$$

is insensitive to the coefficient $s$ of the Bardeen counter-term.

When $\alpha$ or $\bar{\alpha}$ is nonzero, the operator $J_z$ or $J_{\bar{z}}$ is not a Virasoro primary operator, respectively. In a compact reflection-positive CFT, an operator must be either primary or descendent (see for example [46]). Therefore, there must exist an operator $\mathcal{O}$ of dimension $(h, \bar{h}) = (0, 0)$ such that $L_{-1}\mathcal{O} = J_z$ or $\bar{L}_{-1}\mathcal{O} = J_{\bar{z}}$. However, in a compact reflection-positive CFT, the only dimension zero operator is the identity which is annihilated by the Virasoro generators $L_{-1}$ and $\bar{L}_{-1}$. We have learned the following.

**Lesson 3.1.** *A (1+1)d CFT with mixed U(1)-gravitational anomaly cannot be compact and reflection-positive.*

---

[13]In particular, $[L_0, J_0] = \alpha$. In the vertex operator algebra (VOA) language, $[L_1, J(0)] \neq 0$ means that the VOA is "not of strong CFT type". For a strongly rational holomorphic VOA (which requires it to be of strong CFT type), it was proven by [45] that the central charge must be a multiple of 8.

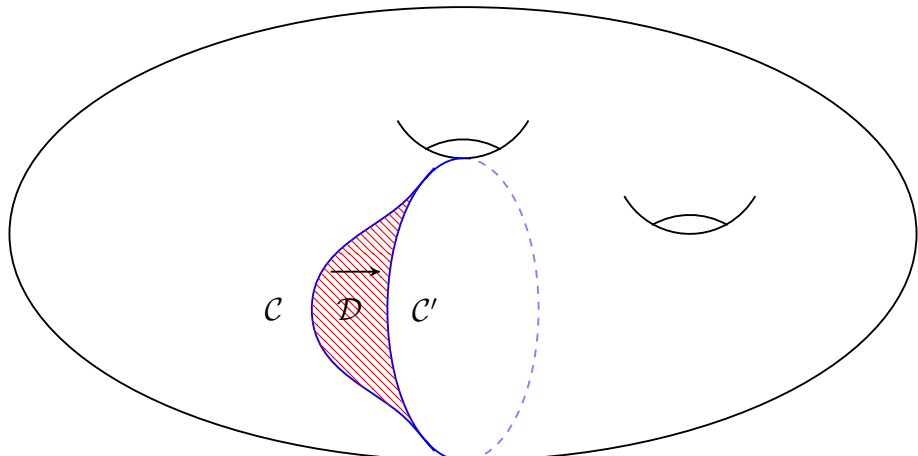

Figure 2: Deforming a symmetry defect line from the curve $\mathcal{C}$ across the domain $\mathcal{D}$ to the new curve $\mathcal{C}'$.

## 3.2 Topological defects and isotopy anomaly

An invertible topological defect line (TDL) can be constructed from a covariant current $\mathcal{J}_\mu$ that is conserved up to covariant anomalies,

$$\mathcal{L}_\eta(\mathcal{C}) \;=\; : \exp\left[i\eta \oint_{\mathcal{C}} ds\, n_\mu \mathcal{J}^\mu\right] :, \tag{3.22}$$

where $n_\mu$ is the normal vector to the curve $\mathcal{C}$. The defect is topological up to an isotopy anomaly: When the curve $\mathcal{C}$ is deformed across a domain $\mathcal{D}$, as shown in Figure 2, the defect $\mathcal{L}_\eta$ is modified by a phase factor determined by the divergence theorem,

$$: \exp\left[i\eta \oint_{\partial\mathcal{D}} ds\, n_\mu \mathcal{J}^\mu\right] : \;=\; : \exp\left[i\eta \int_{\mathcal{D}} d^2x \sqrt{g}\nabla_\mu \mathcal{J}^\mu\right] : \;=\; \exp\left[i\eta\kappa_{FR} \int_{\mathcal{D}} d^2x \sqrt{g}R\right]. \tag{3.23}$$

The isotopy anomaly generalizes the mixed gravitational anomaly to discrete groups and non-invertible topological defects [47]. For discrete groups, there is no analog of Lesson 3.1. In particular, an anomalous $\mathbb{Z}_2$ symmetry defect line in compact reflection-positive CFTs has isotopy anomaly. Note that a defect line defined using the consistent current $J_\mu$ is not topological. Even on flat space, its anomalous conservation depends on the choice of coordinate system. The consistent and covariant currents agree only in Cartesian coordinates on flat space.

### Isotopy anomaly as contact term

On flat space, the isotopy anomaly can be detected by the contact terms in the OPE between the stress tensor and the symmetry defect $\mathcal{L}_\eta$. Using the two-point functions (3.17) and

(3.19), we find

$$\langle T_{zz}(z,\bar{z})\mathcal{L}_\eta\rangle = \eta \left\langle : \oint_{\mathcal{C}} \left[ \frac{\alpha}{(z-w)^3}dw - 2\pi\beta\partial_z\delta^{(2)}(z-w,\bar{z}-\bar{w})d\bar{w} \right]\mathcal{L}_\eta : \right\rangle$$
$$= -i\pi(\alpha+2\beta)\eta\partial_z^2\theta(z\in D)\langle\mathcal{L}_\eta\rangle , \tag{3.24}$$

where we have assumed that TDL $\mathcal{L}_\eta$ is located on the boundary of a compact region $D$, i.e. $\mathcal{C} = \partial D$. Similarly, we also have

$$\langle T_{\bar{z}\bar{z}}(z,\bar{z})\mathcal{L}_\eta\rangle = -i\pi(\bar{\alpha}+2\bar{\beta})\alpha\partial_{\bar{z}}^2\theta(z\in D)\langle\mathcal{L}_\eta\rangle ,$$
$$\langle T_{z\bar{z}}(z,\bar{z})\mathcal{L}_\eta\rangle = -2\pi i(\gamma+\bar{\gamma})\alpha\partial_z\partial_{\bar{z}}\theta(z\in D)\langle\mathcal{L}_\eta\rangle . \tag{3.25}$$

## 3.3  Periodicity anomaly

On flat space, for every each $\eta \in \mathbb{Z}$, the defect $\mathcal{L}_\eta$ commutes with all local operators and is therefore identified with the trivial line, reflecting the periodicity of U(1). On curved space, this family of lines differ by their isotopy anomaly, and the periodicity of U(1) is ruined.[14] A remedy is to modify the topological defect by a local improvement term[15]

$$\widetilde{\mathcal{L}}_\eta(\mathcal{C}) = \mathcal{L}_\eta(\mathcal{C}) \exp\left[ -i\eta\kappa_{FR} \oint_{\mathcal{C}} ds\, K \right] , \tag{3.26}$$

such that the isotopy anomaly is precisely canceled via the Gauss-Bonnet theorem. However, only when $\kappa_{FR} \in \frac{\mathbb{Z}}{2}$ does this fully restore the periodicity of U(1). Otherwise, the distinction between $\widetilde{\mathcal{L}}_{\eta=0}$ and $\widetilde{\mathcal{L}}_{\eta=1}$ can be detected by the loop expectation value on the plane,[16]

$$\langle\widetilde{\mathcal{L}}_\eta(\mathcal{C})\rangle_{\mathbb{R}^2} = \exp\left[ -4\pi i\eta\kappa_{FR} \right] . \tag{3.28}$$

The phase signals a *periodicity anomaly*, analogous to the *orientation reversal anomaly* of a $\mathbb{Z}_2$ symmetry defect line [47]. There, if one insists on cancelling the isotopy anomaly of the anomalous $\mathbb{Z}_2$ symmetry defect line, then orientation reversal (which represents the group inverse operation) turns the $\mathbb{Z}_2$ symmetry defect line into one with a different extrinsic curvature improvement term. Here, the action of $\eta \to \eta+1$ changes the extrinsic curvature

---

[14]In particular, a point of localized curvature can carry an arbitrary real amount of charge.

[15]In [47], this term was called an extrinsic curvature "counter-term" by the present authors. However, from a purely $(1+1)d$ point of view, it is more appropriately regarded as an improvement term for a defect operator.

[16]The loop expectation value of a TDL $\mathcal{L}$ on the plane was denoted by $R(\mathcal{L})$ in [47]. If $\mathcal{C}$ is the unit circle on flat space, then

$$\oint_{\mathcal{C}} ds\, K = 4\pi . \tag{3.27}$$

improvement term. If the quantization condition $\kappa_{FR} \in \frac{\mathbb{Z}}{4}$ in (3.12) obtained from inflow considerations is universally true, then the anomalous phase in (3.26) is at most a sign.

Let us compare the merits of $\mathcal{L}_\eta$ and $\widetilde{\mathcal{L}}_\eta$. If the mixed gravitational anomaly is such that $\kappa_{FR} \in \frac{\mathbb{Z}}{2}$, then $\widetilde{\mathcal{L}}_\eta$ implements the same U(1) symmetry action as $\mathcal{L}_\eta$ (without periodicity anomaly) on flat space, and is free of isotopy anomaly on curved manifolds, unlike $\mathcal{L}_\eta$. Hence $\widetilde{\mathcal{L}}_\eta$ is in all respects better than $\mathcal{L}_\eta$. However, if $\kappa_{FR} \notin \frac{\mathbb{Z}}{2}$, then we are faced with a dilemma.

**Lesson 3.2.** *If the mixed gravitational anomaly is such that $\kappa_{FR} \notin \frac{\mathbb{Z}}{2}$, then*

1. *The topological defect line $\mathcal{L}_\eta$ has no periodicity anomaly on flat space, but has isotopy anomaly on curved background.*

2. *The topological defect line $\widetilde{\mathcal{L}}_\eta$ has periodicity anomaly on flat space, but is free from isotopy anomaly on curved background.*

# 4   Comments on the anomaly of finite subgroups

The quantization of the pure anomaly coefficient $\kappa_{F^2} = 1 \notin 2\mathbb{Z}$ for the U(1) internal symmetry (quantized such that the local operators span integer charges) makes the mapping of the anomaly to discrete subgroups subtle and confusing. When $\kappa_{F^2} \in 2\mathbb{Z}$, the $\mathbb{Z}_N$ subgroup of the U(1) has anomaly

$$\frac{\kappa_{F^2}}{2} \mod N \in H^3(\mathbb{Z}_N, \mathrm{U}(1)). \tag{4.1}$$

However, when $\kappa_{F^2} \in 2\mathbb{Z} + 1$, the anomaly of the $\mathbb{Z}_N$ subgroup does not fall into the classification by group cohomology.

The source of this confusion is the ambiguity in the embedding of background $\mathbb{Z}_N$ gauge configurations (configurations of $\mathbb{Z}_N$ symmetry defect lines described by a $\mathbb{Z}_N$ fusion category) into background U(1) gauge configurations, and in fact some choices of the embedding can be inconsistent, as illustrated by the following example. Consider a $\mathbb{Z}_3$ background gauge configuration described by three identical parallel $\mathbb{Z}_3$ symmetry defect lines wrapped around the time direction on a torus. Naively, this $\mathbb{Z}_3$ background gauge configuration can be embedded into any of the infinitely many U(1) gauge configurations

$$A = 2\pi \left[ \left( \frac{1}{3} + n_1 \right) \delta(x) + \left( \frac{1}{3} + n_2 \right) \delta(x - \frac{2\pi}{3}) + \left( \frac{1}{3} + n_3 \right) \delta(x - \frac{4\pi}{3}) \right], \tag{4.2}$$

where $x$ is the spatial direction and $n_i \in \mathbb{Z}$ parametrize the ambiguity in the embedding. These U(1) background gauge configurations are pure gauges with winding number $n_1 + n_2 + n_3 + 1$ around the spatial circle, *i.e.*

$$A = d\lambda, \quad \lambda = 2\pi \left[ \left( \frac{1}{3} + n_1 \right) \theta(x) + \left( \frac{1}{3} + n_2 \right) \theta(x - \frac{2\pi}{3}) + \left( \frac{1}{3} + n_3 \right) \theta(x - \frac{4\pi}{3}) \right], \tag{4.3}$$

and are thus related to the trivial configuration by a large background gauge transformation, which produces an anomalous phase $\pi\kappa_{F^2}(n_1 + n_2 + n_3 + 1)$ via the $\theta(\vec{m}[\lambda])$ term in (2.44). When $\kappa_{F^2} \in 2\mathbb{Z}+1$, the anomalous phase depends on $n_1 + n_2 + n_3 \mod 2$ of the embedding. However, according to the fusion axiom [47], three identical parallel $\mathbb{Z}_3$ symmetry defect lines on a torus fuse to the trivial line with no anomalous phase, so consistency with the axiom imposes the condition

$$n_1 + n_2 + n_3 + 1 \in 2\mathbb{Z}. \tag{4.4}$$

Let us recap the relation between configurations of topological defect lines and background U(1) gauge configurations. On the one hand, the configuration of three identical parallel $\mathbb{Z}_3$ symmetry defect lines (embedded in U(1)) could correspond to any of the background gauge configurations (4.2) with $n_1 + n_2 + n_3 + 1 \in 2\mathbb{Z}$. On the other hand, background U(1) gauge configurations with $n_1 + n_2 + n_3 \in 2\mathbb{Z}$ cannot be represented by a configuration of topological defect lines.

# 5 Holomorphic $bc$ ghost system

The anomalies of the previous sections will now find life in a specific theory — the holomorphic $bc$ ghost system, which is a CFT of anti-commuting complex free fields $b$ and $c$, with weights

$$h_b = \lambda, \quad h_c = 1 - \lambda \tag{5.1}$$

and OPE

$$b(z)c(0) \sim \frac{1}{z}. \tag{5.2}$$

The stress tensor and the U(1) current for the ghost number symmetry that assigns charges $\pm 1$ to $c$ and $b$ are

$$T_{zz} = (1 - \lambda) : (\partial b)c : -\lambda : b\partial c :, \quad J_z =: bc :. \tag{5.3}$$

The anomalies coefficients, computed from the $TT$, $jj$, and $Tj$ OPEs, are

$$c_- = \kappa_{R^2} = 1 - 3(2\lambda - 1)^2, \quad k_- = \kappa_{F^2} = 1, \quad \kappa_{FR} = \frac{2\lambda - 1}{4}. \tag{5.4}$$

To be a non-spin CFT, the spins of $b$ and $c$ must be integers, hence

$$\lambda \in \mathbb{Z}. \tag{5.5}$$

The U(1) anomaly coefficient $\kappa_F^2$ is not an even integer, so it cannot be canceled by a bulk $(2+1)d$ U(1) Chern-Simons. Likewise, the gravitational anomaly coefficient $\kappa_{R^2} \in 8\mathbb{Z}-2$ is an even integer but not a multiple of eight. Hence, the gravitational anomaly cannot be canceled by a bulk $(2+1)d$ gravitational Chern-Simons. They do, however, satisfy and saturate the quantization conditions derived form the finite Wess-Zumino consistency conditions.

## Torus one-point function of the ghost number current

Let us consider the holomorphic $bc$ ghost system on a flat torus with complex moduli $\tau$, with periodic boundary conditions around both the space and Euclidean time cycles. The torus partition function vanishes due to the zero modes of the $b$ and $c$ fields. Consider instead the torus one-point function of the current $J_z$,

$$G(\tau, \bar{\tau}) = \langle J_z(0) \rangle_{T^2_\tau} = \eta(\tau)^2. \tag{5.6}$$

Under modular $S$ and $T$ transformations, the anomalous phases are

$$\theta_S = \frac{3\pi}{2}, \quad \theta_T = \frac{\pi}{6}, \tag{5.7}$$

which satisfy the quantization condition (2.40) for $\ell = 1$.

## Flavored torus partition function

Consider a flat torus with metric[17]

$$ds^2 = (d\sigma^1)^2 + (d\sigma^2)^2, \quad \sigma^1 \cong \sigma^1 + 2\pi\mathbb{Z}, \quad \sigma^1 \cong \sigma^1 + 2\pi\mathbb{Z}, \tag{5.8}$$

where $\tau = \tau_1 + \tau_2$ is the complex moduli, and let us compute the partition function of the $bc$ system on this torus with constant background gauge field

$$A = A_1 d\sigma^1 + A_2 d\sigma^2. \tag{5.9}$$

A natural thing to evaluate is the trace

$$Z_{\mathrm{H}}(\tau, z) = \mathrm{Tr}\left(q^{L_0 - \frac{c}{24}} e^{2\pi i(z - \frac{1}{2})J_0}\right) = \frac{\theta_1(\tau|z)}{\eta(\tau)}, \tag{5.10}$$

where the chemical potential $z$ is related to the constant background gauge field $A$ by

$$z = -i\tau_2(A_1 + iA_2). \tag{5.11}$$

However, $Z_{\mathrm{H}}(\tau, z)$ does not satisfy the transformation law (1.1). The resolution is an extra term $B$ that comes from carefully taking the Legendre transformation that relates the Lagrangian to the Hamiltonian [48, 49], resulting in

$$Z(\tau, \bar{\tau}, z, \bar{z}) = Z_{\mathrm{H}}(\tau, z)\, e^{\pi B(\tau, \bar{\tau}, z, \bar{z})}. \tag{5.12}$$

---

[17]This is a different coordinate system from (2.24).

The function $B$ a quadratic function in $z$ and $\bar{z}$ (by nature of the Legendre transform), that vanishes when $A_1 = 0$, and transforms under $\mathrm{SL}(2, \mathbb{Z})$ as

$$B\left(\frac{a\tau + b}{c\tau + d}, \frac{a\bar{\tau} + b}{c\bar{\tau} + d}, \frac{z}{c\tau + d}, \frac{\bar{z}}{c\bar{\tau} + d}\right) = B(\tau, \bar{\tau}, z, \bar{z}) - \frac{icz^2}{c\tau + d} \tag{5.13}$$

so that the flavored partition function $Z(\tau, \bar{\tau}, z, \bar{z})$ is invariant under $\mathrm{SL}(2, \mathbb{Z})$ up to anomalous phases. It is fixed to be

$$B(\tau, \bar{\tau}, z, \bar{z}) = \frac{z(z - \bar{z})}{2\tau_2}. \tag{5.14}$$

Under the modular $S$ and $T$ transformations, the flavored partition function transforms as

$$Z(\tau + 1, \bar{\tau} + 1, z, \bar{z}) = e^{\frac{\pi i}{6}} Z(\tau, \bar{\tau}, z, \bar{z}),$$
$$Z\left(-\frac{1}{\tau}, -\frac{1}{\bar{\tau}}, \frac{z}{\tau}, \frac{\bar{z}}{\tau}\right) = e^{\frac{3\pi i}{2}} Z(\tau, \bar{\tau}, z, \bar{z}), \tag{5.15}$$

which agrees with the anomalous phases (5.7). Under a large gauge transformation

$$A \to A + d\lambda, \quad \lambda = m\left(\sigma^1 - \frac{\tau_1}{\tau_2}\sigma_2\right) + n\frac{\sigma^2}{\tau_2}, \tag{5.16}$$

the flavored partition function transforms as

$$Z(\tau, \bar{\tau}, A + d\lambda) = Z(\tau, \bar{\tau}, A) \exp\left[-\pi i(m\tau_2 A_2 - (n - m\tau_1)A_1) - (mn + m + n)\pi i\right]$$
$$= Z(\tau, \bar{\tau}, A) \exp\left(-\frac{i}{4\pi}\int d\lambda A - (mn + m + n)\pi i\right), \tag{5.17}$$

which also agrees with (2.44) with $\kappa_{F^2} = 1$.

# 6 Concluding remarks

Starting with the finite Wess-Zumino consistency condition (1.3), we derived quantization conditions on the pure gravitational and U(1) anomaly coefficients $\kappa_{R^2}$ and $\kappa_{F^2}$ in $(1+1)d$ non-spin quantum field theory. The quantization conditions turned out to be weaker than those predicted by the inflow of properly quantized classical Chern-Simons actions. We also examined the mixed U(1)-gravitational anomaly, proposed an inflow mechanism, and from inflow derived a quantization condition on $\kappa_{FR}$. It may be possible to derive a quantization condition on $\kappa_{FR}$ from the finite Wess-Zumino consistency alone without invoking inflow, but this requires going beyond the flat torus background to *e.g.* a genus-two Riemann surface, and is beyond the scope of this paper. The quantization conditions are summarized

|  | Inflow | fWZ |
|---|---|---|
| $\kappa_{R^2} = c_-$ | $8\mathbb{Z}$ | $2\mathbb{Z}$ |
| $\kappa_{F^2} = k_-$ | $2\mathbb{Z}$ | $\mathbb{Z}$ |
| $\kappa_{FR}$ | $\frac{1}{4}\mathbb{Z}$ | ? |

Table 1: Quantization of anomaly coefficients predicted by inflow of classical Chern-Simons actions versus the finite Wess-Zumino consistency condition (1.3).

in Table 1. A survey of the holomorphic $bc$ ghost system found the theory to realize the minimal quantization condition for all three anomalies.

We called an anomaly exotic if

$$\kappa_{R^2} \notin 8\mathbb{Z} \quad \text{or} \quad \kappa_{F^2} \notin 2\mathbb{Z} \quad \text{or} \quad \kappa_{FR} \neq 0 \,, \tag{6.1}$$

and proved that certain exotic anomalies cannot be realized in any compact reflection-positive non-spin conformal field theory. The lack of reflective-positivity is no reason to dismiss these exotic anomalies.[18] Besides the central role Faddeev-Popov ghosts play in gauge theories, ghost fields also appear in interesting holographic contexts, including a purported holographic dual of dS$_4$ higher spin gravity [53–55], and supergroup gauge theories [56–58].

The mixed gravitational anomaly discussed in Section 3 has a natural even $D$-dimensional generalization, described by an anomalous phase $\mathcal{A}_\lambda$ that involves the $D$-dimensional Euler form $E_D$,

$$\mathcal{A}_\lambda = \kappa_{FR} \int_{\mathcal{M}_D} \lambda\, E_D, \quad E_D = \frac{1}{(2\pi)^{\frac{D}{2}}} \varepsilon^{a_1, \cdots, a_D} R_{a_1 a_2} \wedge \cdots \wedge R_{a_{D-1} a_D} \,, \tag{6.2}$$

where we ignored the ambiguity from Bardeen counter-terms. An inflow mechanism of this anomaly involves a mixed classical Chern-Simons action of a background U(1) gauge field and a background SO($D$) gauge field that matches with the $D$-dimensional spin-connection by a boundary matching condition analogous to (3.7). On product manifolds $\mathcal{M}_D \times [0,1)$ with a distinguished time direction, a higher-dimensional generalization of the Wen-Zee topological term [33, 34] can also provide the inflow. Further study of the higher-dimensional mixed gravitational anomaly is left for future work.

---

[18]Lattice models at criticality need not be reflection-positive. Non-reflection-positive CFTs are known to be important landmarks in RG space that in fact influence reflection-positive RG flows [50, 51]. Quantum field theory realizing non-integer "$O(N)$" symmetry is necessarily non-reflection-positive [52].

# Acknowledgements

We are grateful to Po-Shen Hsin, Chao-Ming Jian, Shu-Heng Shao, Ryan Thorngren, Yifan Wang and Xiao-Gang Wen for helpful discussions and comments. CC thanks the hospitality of National Taiwan University. YL is supported by the Sherman Fairchild Foundation, by the U.S. Department of Energy, Office of Science, Office of High Energy Physics, under Award Number DE-SC0011632, and by the Simons Collaboration Grant on the Non-Perturbative Bootstrap.

# A    Bardeen-Zumino counter-terms

In this appendix, we review the pure Bardeen-Zumino counter-term of [37], and construct a mixed Bardeen-Zumino counter-term for the mixed U(1)-gravitational anomaly of Section 3.

## A.1    Pure gravitational

Let us treat the vielbein $e^a{}_\mu$ as a matrix and denote it by $E$. The two-dimensional pure Bardeen-Zumino term is

$$S_{\text{BZ}} = -\frac{i}{2\pi} \int_{\mathcal{M}_2} \int_0^1 dt\, \frac{\kappa_{R^2}}{48} \text{tr}\, (Hd\Gamma_t) = -\frac{i}{2\pi} \int_{\mathcal{M}_2} \int_0^1 d\tau\, \frac{\kappa_{R^2}}{48} \text{tr}\, (Hd\omega_\tau)\,, \tag{A.1}$$

where the $H$ is

$$E = e^H\,, \tag{A.2}$$

and the $\Gamma_t$ and the $\omega_t$ are defined by

$$\Gamma_t = E^t \Gamma E^{-t} + E^t dE^{-t} = E^{-1+t}\omega E^{1-t} + E^{-1+t}dE^{1-t} = \omega_\tau\,, \tag{A.3}$$

where $\tau = 1 - t$. The matrix valued Christoffel one-form $\Gamma$ and spin connection $\omega$ are defined by

$$\Gamma^\mu{}_\nu \equiv \Gamma^\mu{}_{\nu\rho} dx^\rho\,, \quad \omega^a{}_b \equiv \omega^a{}_{b\mu} dx^\mu\,. \tag{A.4}$$

In the matrix notation, diffeomorphisms and local frame rotations act on the viebein $E$, Christoffel one-form $\Gamma$, and spin connection $\omega$ as

$$\begin{aligned}
\delta_\xi E &= (\mathcal{L}_\xi + T_\Lambda)E\,, & \delta_\xi \Gamma &= (\mathcal{L}_\xi + T_\Lambda)\Gamma\,, \\
T_\Lambda E &\equiv E\Lambda\,, & T_\Lambda \Gamma &\equiv d\Lambda + [\Gamma, \Lambda]\,, \\
\delta_\theta E &= -\theta E\,, & \delta_\theta \omega &= d\theta + [\omega, \theta]\,.
\end{aligned} \tag{A.5}$$

where $\mathcal{L}_\xi$ is the Lie derivative.[19] The gauge parameter $\Lambda$ is related to the diffeomorphism parameter $\xi$ by

$$\Lambda^\rho{}_\mu = \partial_\mu \xi^\rho. \tag{A.7}$$

The $\Gamma_t$ transforms under $T_\Lambda$ as

$$T_\Lambda \Gamma_t = d\Lambda_t + [\Gamma_t, \Lambda_t] \equiv T_{\Lambda_t}\Gamma_t, \quad \Lambda_t \equiv E^t \Lambda E^{-t} + E^t(T_\Lambda E^{-t}). \tag{A.8}$$

We also have the identities

$$\begin{aligned}
\frac{\partial \Lambda_t}{\partial t} &= [H, \Lambda_t] - T_\Lambda H, \\
\frac{\partial \Gamma_t}{\partial t} &= -dH + [H, \Gamma_t].
\end{aligned} \tag{A.9}$$

Using the above, we compute the diffeomorprhism variation of the Bardeen-Zumino action to be

$$\delta_\xi S_{\mathrm{BZ}} = -\frac{i}{2\pi} \int_{\mathcal{M}_2} \frac{\kappa_{R^2}}{48} \mathrm{tr}\,(\Lambda d\Gamma). \tag{A.10}$$

By a similar computation, we find

$$\delta_\theta S_{\mathrm{BZ}} = \frac{i}{2\pi} \int_{\mathcal{M}_2} \frac{\kappa_{R^2}}{48} \mathrm{tr}\,(\theta d\omega). \tag{A.11}$$

Hence, adding the Bardeen-Zumino counter-term $S_{\mathrm{BZ}}$ to the effective action $W[e, A]$, we cancel the pure frame rotation anomaly, *i.e.* $\mathcal{A}_\theta$ in (2.8), while introducing a pure diffeomorphism anomaly (2.17).

## A.2 Mixed gravitational

We introduce a mixed Bardeen-Zumino action

$$S_{\mathrm{BZ}}^{\mathrm{mixed}} = -\frac{i}{2\pi} \int_0^1 dt \int_{\mathcal{M}_2} \left(\frac{\kappa_{FR}}{2} + s\right) \mathrm{tr}\,(Hd(\mathcal{E}_t \mathbb{A})). \tag{A.12}$$

The matrix $\mathcal{E}_t$ is defined by

$$\mathcal{E}_t = E^t \mathcal{E} E^{-t}, \tag{A.13}$$

where the matrix $\mathcal{E}$ is the $(1+1)d$ Levi-Civita tensor $\varepsilon^\mu{}_\nu$. Note that the matrix $\mathscr{E} \equiv \mathcal{E}_1$ is the Levi-Civita symbol $\varepsilon^a{}_b$ with local Lorentz indices. The $\mathcal{E}_t$ transforms under $T_\Lambda$ as

$$T_\Lambda \mathcal{E}_t = [\mathcal{E}_t, \Lambda_t]. \tag{A.14}$$

---

[19] An useful identity between the Lie derivative $\mathcal{L}_\xi$, exterior derivative $d$ and interior product $\iota_\xi$ is

$$\mathcal{L}_\xi = d\iota_\xi + \iota_\xi d. \tag{A.6}$$

We also have identity

$$\frac{\partial \mathcal{E}_t}{\partial t} = H E^t \mathcal{E} E^{-t} - E^t \mathcal{E} E^{-t} H = [H, \mathcal{E}_t] \,. \tag{A.15}$$

Using the above, we obtain the diffeomorprhism variation of the mixed Bardeen-Zumino action to be

$$\delta_\xi S_{\text{BZ}}^{\text{mixed}} = -\frac{i}{2\pi} \int_{\mathcal{M}_2} \left( \frac{\kappa_{FR}}{2} + s \right) \text{tr} \left( \Lambda d(\mathcal{E} A) \right) \,. \tag{A.16}$$

Similarly, we have the variation of the mixed Bardeen-Zumino term under local frame rotations,

$$\delta_\theta S_{\text{BZ}}^{\text{mixed}} = \frac{i}{2\pi} \int_{\mathcal{M}_2} \left( \frac{\kappa_{FR}}{2} + s \right) \text{tr} \left( \theta \mathcal{E} \right) dA \,. \tag{A.17}$$

To derive the variation of the mixed Bardeen-Zumino term under the background U(1) gauge transformation, let us first rewrite the mixed Bardeen-Zumino term by integrating out the auxiliary variable $t$ in (A.12) as

$$S_{\text{BZ}}^{\text{mixed}} = -\frac{i}{2\pi} \int_{\mathcal{M}_2} \left( \frac{\kappa_{FR}}{2} + s \right) \text{tr} \left( \omega \mathcal{E} - \Gamma \mathcal{E} \right) A. \tag{A.18}$$

Under background U(1) gauge transformations, the mixed Bardeen-Zumino term becomes

$$\delta_\lambda S_{\text{BZ}}^{\text{mixed}} = -\frac{i}{2\pi} \int_{\mathcal{M}_2} \left( \frac{\kappa_{FR}}{2} + s \right) \lambda \text{tr} \left( \mathcal{E} d\omega - d(\Gamma \mathcal{E}) \right). \tag{A.19}$$

# B  No covariant stress tensor for mixed anomaly

Let us examine the possibility of improving the stress tensor such that the mixed gravitational anomaly becomes covariant. The most general improvement terms linear in derivatives and linear in $A$ come in two forms, $\partial A$ and $\Gamma A$. For the first form, it is clear that there are two possibilities

$$\partial^{(\mu} A^{\nu)} \,, \quad g^{\mu\nu} \partial^\sigma A_\sigma \,. \tag{B.1}$$

For the second form, if $A$ takes a $\mu, \nu$ index, then we have

$$g^{\rho\sigma} \Gamma^\mu_{\rho\sigma} A^\nu \,, \quad g^{\mu\rho} \Gamma^\sigma_{\rho\sigma} A^\nu \,, \tag{B.2}$$

and if $A$ takes a dummy index that is contracted, then we have

$$g^{\mu\rho} \Gamma^\nu_{\rho\sigma} A^\sigma \,, \quad g^{\mu\rho} g^{\nu\sigma} \Gamma^\tau_{\rho\sigma} A_\tau \,, \quad g^{\mu\nu} \Gamma^\rho_{\rho\sigma} A^\sigma \,, \quad g^{\mu\nu} g^{\rho\sigma} \Gamma^\tau_{\rho\sigma} A_\tau \,. \tag{B.3}$$

Hence, the most general improvement takes the form

$$\begin{aligned}
Y_2^{\mu\nu} &= c_1 \Gamma^{(\mu\nu)\rho} A_\rho + c_2 \Gamma^{\rho\mu\nu} A_\rho + c_3 \Gamma^{(\underline{\mu}\rho\sigma} g_{\rho\sigma} A^{\nu)} + c_4 g_{\rho\sigma} \Gamma^{\rho\sigma(\mu} A^{\nu)} \\
&\quad + c_5 g^{\mu\nu} \Gamma^\sigma{}_{\sigma\rho} A^\rho + c_6 g^{\mu\nu} g_{\sigma\lambda} \Gamma^{\rho\sigma\lambda} A_\rho + c_7 g^{(\underline{\mu}\rho} g^{\nu)\sigma} \partial_\rho A_\sigma + c_8 g^{\mu\nu} g^{\rho\sigma} \partial_\rho A_\sigma \,.
\end{aligned} \tag{B.4}$$

The only possible covariant form of the conservation equation is

$$\langle \nabla_\mu \mathcal{T}^{\mu\nu}(x) \rangle \supset \nabla_\mu F^{\mu\nu} \,. \tag{B.5}$$

Using the MathGR package [59], it is straightforward to evaluate $\nabla_\mu Y_2^{\mu\nu}$, $\nabla_\mu F^{\mu\nu}$ and the consistent mixed gravitational anomaly in $\nabla_\mu T^{\mu\nu}$ in conformal gauge. The results can be decomposed with respect to a basis (with the overall conformal factor $e^{-4w}$ stripped off)

$$\begin{aligned}
&(A\partial)w\partial_\nu w \,, \quad A_\nu \partial^2 w \,, \quad (A\partial)\partial_\nu w \,, \quad \partial_\nu w(\partial A) \,, \\
&(\partial_\nu A_\rho)\partial_\rho w \,, \quad \partial_\nu(\partial A) \,, \quad \partial_\rho w \partial_\rho A_\nu \,, \quad \partial^2 A_\nu \,,
\end{aligned} \tag{B.6}$$

where $\partial A = \partial_\rho A_\rho$ and $A\partial = A_\rho \partial_\rho$. In this basis, the eight terms in $\nabla_\mu Y_2^{\mu\nu}$ can be represented by a coefficient matrix

$$\begin{pmatrix}
2 & -1 & -1 & -2 & 0 & \frac{1}{2} & -1 & \frac{1}{2} \\
4 & 0 & -2 & -2 & -2 & 1 & 0 & 0 \\
0 & 0 & 0 & 0 & 0 & 0 & 0 & 0 \\
-2 & 1 & 1 & 1 & 0 & 0 & 1 & 0 \\
-2 & 0 & 1 & 0 & 1 & 0 & 0 & 0 \\
0 & 1 & 0 & 1 & -1 & 0 & 1 & 0 \\
-4 & 0 & 2 & 0 & 2 & 0 & 0 & 0 \\
0 & 0 & 0 & 0 & 0 & 0 & 0 & 0
\end{pmatrix} \,. \tag{B.7}$$

The covariant anomaly $\nabla_\mu F^{\mu\nu}$ is represented by

$$\begin{pmatrix} 0 & 0 & 0 & 0 & -2 & 1 & 2 & -1 \end{pmatrix} \,, \tag{B.8}$$

and the consistent anomaly in $\nabla_\mu T^{\mu\nu}$ is represented by

$$\begin{pmatrix} 0 & 0 & 0 & 0 & 0 & -1 & 0 & 1 \end{pmatrix} \,. \tag{B.9}$$

No combination of (B.8) with the rows of (B.7) produces (B.9). Hence, no covariant stress tensor exists.

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
