# Peer review of "On Exotic Consistent Anomalies in (1+1)$d$: A Ghost Story"

_SciPost Physics_

## Round 3 · Referee Report · Anonymous · 2021-1-16

Strengths

1- Detailed analysis of anomalies in 1+1-dimensional non-unitary theories.
2- Relatively not-well studied but important subject.
3- The logic and derivation are clear.

Report

The authors studied the perturbative anomaly in possibly non-unitary 1+1-dimensional QFTs with U(1) symmetry.
The careful analysis reveals the properties that are previously not appreciated and opens the door to study more general anomalies in non-unitary QFTs. Therefore I recommend publishing the manuscript in SciPost Physics.

A few questions/comments:
1- Is there any relation among the anomaly coefficients? Is there a linear combination of them that should vanish modulo something, or are they independent?
2- I could not understand what is the conclusion in section 4. When $\kappa_F$ is odd, the topological lines describing the $Z_3$ lines cannot satisfy some of the axioms of fusion category?
3- (Just a comment) The inflow action involving the Euler characteristic exemplifies that the classification of the anomalies in non-unitary QFTs cannot be given by a *stable* homotopy theory. Here, to write down the Euler characteristic, the tangent bundle of $M_2$ had to be extended into the bulk as a $SO(2)$ bundle, while in a unitary theory the anomaly does not care much about $d$ of $SO(d)$. It would be nice to find and study the unstable homotopy that could describe these anomalies.

Requested changes

Just typos
1- Above (5.4), "anomalies coefficients" -> "anomaly coefficients"
2- Below (6.1), "reflective-positivity" -> "reflection-positivity"

  • validity: top
  • significance: high
  • originality: high
  • clarity: top
  • formatting: -
  • grammar: -

Author:  Ying-Hsuan Lin  on 2021-02-11  [id 1225]

(in reply to Report 1 on 2021-01-16)

We thank the referee for the comments, and have made corrections and adjustments accordingly in our updated version. Our responses are below:

1- It is an interesting question. We did suspect that $\kappa_{F^2}$ might be odd iff $4\kappa_{FR}$ is odd, but we could not prove it.

2- That’s right. When $\kappa$ is odd, the anomalous phase depends on extra information that is not captured in the framework of background $\mathbb{Z}_N$ gauge transformations (or equivalently manipulations in $\mathbb{Z}_N$ fusion categories). We rewrote that section to clarify the discussion.

3- We thank the referee for this insightful comment. While the subject of stable homotopy is beyond the expertise of the authors, we added footnote 21 and thanked the referee.

---

## Editorial Decision

resubmitted